# Long-term music instruction is partially associated with the development of socioemotional skills

**Jed Villanueva**[1], **Beatriz Ilari**[2], **Assal Habibi**[1]*

**1** Brain and Creativity Institute, Dornsife College of Letters Arts and Sciences, University of Southern California, Los Angeles, CA, United States of America, **2** Department of Music Teaching & Learning, Thornton School of Music, University of Southern California, Los Angeles, CA, United States of America

* ahabibi@usc.edu

**Data Availability Statement:** All data and code for analysis are available from the Open Science Framework at DOI 10.17605/OSF.IO/Y6DFC, https://osf.io/y6dfc/.

## Abstract

This study aims to investigate the development of pitch-matching, rhythmic entrainment, and socioemotional skills in children who received formal music instruction and other non-music based after school programs. Eighty-three children, averaging 6.81 years old at baseline, were enrolled in either a music, sports, or no after-school program and followed over four years. The music program involved formal and systematic instruction in music theory, instrumental technique, and performance. Most control participants had no music education; however, in some instances, participants received minimal music education at school or at church. Musical development was measured using a pitch-matching and drumming-based rhythmic entrainment task. Sharing behavior was measured using a variation of the dictator game, and empathy was assessed using three different assessments: the Index of Empathy for Children and Adolescence (trait empathy), the Reading the Mind in the Eyes Test (theory of mind), and a Fiction Emotion-Matching task (state empathy). Results revealed no time-related associations in pitch-matching ability; however, formal music instruction improved pitch-matching relative to controls. On the contrary, improvements in rhythmic entrainment were best explained by age-related changes rather than music instruction. This study also found limited support for a positive association between formal music instruction and socioemotional skills. That is, individuals with formal music instruction exhibited improved emotion-matching relative to those with sports training. In terms of general socioemotional development, children's trait-level affective empathy did not improve over time, while sharing, theory of mind, and state empathy did. Additionally, pitch-matching and rhythmic entrainment did not reliably predict any socioemotional measures, with associations being trivial to small. While formal music instruction benefitted pitch-matching ability and emotion-matching to an audiovisual stimulus, it was not a significant predictor of rhythmic entrainment or broader socioemotional development. These findings suggest that the transfer of music training may be most evident in near or similar domains.

**Funding:** The Brain and Music Program (AH) at the Brain and Creativity Institute is supported by the GRoW at Annenberg Foundation (https://growannenberg.org/), the Los Angeles Philharmonic Association (https://www.laphil.com/), the Van Otterloo Family Foundation, and the National Endowment for the Arts (grant 1809970-38-18, https://www.arts.gov/). The funders had no role in study design, data collection and analysis, decision to publish, or preparation of the manuscript. There was no additional external funding received for this study.

**Competing interests:** The authors have declared that no competing interests exist.

# Introduction

Formal music instruction involving the intensive and systematic teaching of music theory, instrumental technique, and performance, has been shown to reliably improve pitch-matching [1–3]—mimicking an external pitch—and rhythmic entrainment [4, 5]—the synchronization of internal rhythms with an external rhythm, two fundamental abilities that shape our experiences with music. In addition, formal music instruction has been shown to aid in socioemotional development, or "the maturation of social and emotional functioning" as defined by Kuther [6]. In line with this definition, we refer to socioemotional skills as traits and capacities which contribute to an individual's ability to maintain social relationships and respond to the emotions of others. Socioemotional skills encompass multiple psychological constructs important for impacting an individuals' overall well-being. Self-esteem, theory of mind, prosociality, perspective taking, and trait and state empathy fit under this umbrella term as they are all integral in assisting individuals engage in positive social interactions. For instance, theory of mind is the ability to recognize that other individuals have unique mental states separate from one's own [7] and is critical in understanding the emotional experience of another person. Likewise, empathy can be categorized into two subcomponents: affective and cognitive [8]. Affective empathy focuses on the ability to recognize and resonate with the emotional state of another person, while cognitive empathy focuses on the ability to understand what others are feeling. In addition, researchers often categorize empathy as either trait or state empathy. Trait empathy describes one's general ability to display empathy, while state empathy describes one's affective reaction given different situations or contexts [9]. Engagement in a formal music training program may then offer a unique opportunity to improve one's capacity for socioemotional skills by enhancing state empathy and encouraging social cohesion.

## Pitch-matching and rhythmic entrainment development

Pitch-matching requires the complex coordination of vocal, motor, and auditory processes; individuals must perceive pitch and temporal information, map this onto motor plans, and continuously regulate their vocal output [10]. Pitch-matching emerges naturally over time [11], with general improvements occurring throughout childhood [12, 13]. Musical engagement, including pitch-matching, performing, listening, and dancing, is predicated on the emergence of rhythmic entrainment, or the spontaneous synchronization of bodily behaviors with external rhythmic processes [14, 15]. As with pitch-matching, the foundations for rhythmic entrainment emerge early in childhood [16]—defined as the period from birth to eight years old [17]—and continue to develop with age [18, 19] and improved motor control [20]. Rhythmic entrainment facilitates the coordination of bodies and voices to the musical beat (i.e., a temporally structured event) through interaction. Thus, scholars often speak of temporal and social aspects of entrainment [21]. Given that collective music making requires humans to synchronize their bodies and voices to the musical beat, formal music instruction is expected to further augment rhythmic entrainment by nurturing one's ability to detect temporal patterns within complex stimuli (i.e., music) and form predictions accordingly. In terms of musical engagement, rhythmic entrainment enables individuals to move in synchrony to the beat of a song and monitor multiple musical events within an auditory scene [22]. This phenomenon can then be observed through physical behaviors such as tapping, head bobbing, and dancing. When applied to social interactions, rhythmic entrainment allows individuals to better synchronize motor and vocal behaviors with another person [23–25].

Beyond their role in music perception, pitch-matching and rhythmic entrainment are vital in shaping our social interactions. For instance, pitch-matching is implicated with establishing social bonds and enabling effective communication through vocal synchrony and emotion

recognition [26]. On the other hand, rhythmic entrainment offers a more direct relationship with social outcomes; the temporal synchronization of physical behaviors with others improves cooperation [27–29], social bonding [30, 31], and the development of a collective identity [32].

## Pitch-matching

Vocal synchrony is the phenomenon by which individuals modulate their speaking patterns to align with their conversation partners. This includes adjustments in pronunciation [33] and syntactic structure [see 34], as well as the modulation of prosodic cues such as speaking rate [35], voice intensity [36], and pitch [37, 38]. Such adaptation in speech encourages interlocutors to adopt the same mental representations, decreasing misunderstandings and improving perceptions of familiarity [37], attractiveness and likability [39, 40], and rapport [41]. For instance, high levels of pitch accommodation during combative conversations (i.e., problem-solving during couples therapy) has been correlated with positive rather than negative attitudes [42]. Similarly, in a study evaluating the performance of therapists during psychological counseling, Imel et al. [38] identified sessions with high levels of empathy to correlate with a greater degrees of vocal pitch synchronization between the therapist and patient. Altogether, this mirroring of vocal behaviors improves successful communication and leads to positive social perceptions of conversation partners. However, the ability to accurately identify prosodic cues varies greatly among individuals, causing widespread interest in determining these contributing factors.

Given evidence that formal music instruction enhances auditory and linguistic processing [see 43], considerable attention has been devoted to evaluating the musician's advantage for prosodic perception. In fact, musicians demonstrate improved perception for prosodic cues such as pitch [44–46] rhythm/timing [47–49], and timbre [50]. Consistent with these findings, musicians generally outperform nonmusicians in identifying emotions in both musical and non-musical stimuli (e.g., [51–54], but see [55]). Notably, a recent study by Greenspon and Montanaro [26] found a significant correlation between pitch-matching and vocal emotion recognition, suggesting that individuals with better pitch-matching ability may be better equipped at understanding the emotions in another's voice. This enhanced auditory processing might then explain why musicians appear to exhibit stronger and more consistent affective responses to musical stimuli (e.g., [56–58]).

## Rhythmic entrainment

As for rhythmic entrainment, the effects of interpersonal synchrony on socioemotional behavior are extensive and can be explained through the blurring of self-other boundaries, and an increased allocation of attentional resources towards group members (see [59]). As such, individuals interacting in time have been found to improve on multiple dimensions of prosocial behavior ([29, 30, 60]), or voluntary actions intended to benefit another individual [61]. For example, Kirschner and Tomasello [28] found that 4-year-old children were more likely to spontaneously help and cooperate with another child after playing children's musical games that involve singing and dancing. Likewise, 8-year-old children engaged in a group singing activity were more likely to cooperate during a prisoner's dilemma game, than children involved in a group art or competitive games activity [27]. However, it seems that engaging in the same behavior may be insufficient in explaining these socioemotional gains. In a study by Hove and Risen [30], college-aged participants completed a task which involved tapping with an experimenter. Once completed, participants were asked to rate the experimenter's likability. Hove and Risen found that the degree of tapping synchrony between participant and

experimenter predicted affiliation ratings, and not just the shared act of tapping together. Similar results were also found in developmental studies, whereby, children's social attitudes improved more when tapping in synchrony with a partner than simply tapping together [62]. Yet this prosocial effect of synchronization accuracy was not found during a spontaneous helping task when an experimenter accidentally dropped some items, or for sharing behaviors, when an experimenter implied, but did not explicitly state, that they'd like the participant to share their resources during a cooperative game [63].

## Socioemotional development

Beyond the near transfer of music-related skills, formal music instruction has been associated with the development of more general aspects of cognition such as speech-in-noise recognition [64], executive functioning [65], language [66], and visuospatial skills [67]. These claims, largely supported by correlational studies, are incapable of establishing a causal relationship between music training and far transfer effects given a lack of random assignment and the absence of a comparable active control group [68, 69]. As such, while many studies might identify a positive association with music training and other cognitive domains, several meta-analyses generally find music training to produce either small [70], or nonexistent [71, 72] transfer effects when accounting for methodological quality.

The benefits of music instruction may instead lie in the social and emotional domains, based on the assumption that music's primary purpose is for social bonding rather than cognitive enhancement. As such, it stands to reason that improvements in pitch-matching ability and rhythmic entrainment would result in an increase in socioemotional skills such as sharing behavior and emotion-recognition ability given their associations with synchronization and emotional prosody. In fact, a recent meta-analysis [73], found 11 out of 17 studies on music training and emotion recognition to have a significant musicians' advantage in emotion recognition to vocal stimuli. However, much of this evidence is once again built on cross-sectional studies necessitating future research with more rigorous study designs controlling for preexisting differences.

In general, large age-related changes in socioemotional skills can be observed early during infancy. This transition between infancy and toddlerhood sees a notable increase in related capacities such as social-cognitive awareness, empathic responding, and self-other distinction. For instance, rudimentary forms of prosocial behavior and expectations for helping and cooperation have been observed in 9- to 18-month-olds [74]. As babies transition into early childhood (e.g., ages 3 and 4), their comprehension of social situations begins to incorporate more sophisticated elements such as an understanding that other individuals possess their own thoughts, beliefs, and emotions or theory of mind [75]. Similarly, middle childhood (e.g., 5- to 10-years-old) has been shown to coincide with substantial gains in the empathy domain, such as the development of an advanced theory of mind, or an understanding that other individuals may hold inaccurate beliefs about the beliefs of a third party [76].

In research on the neural basis of social behavior, adolescents compared to adults have exhibited decreased functional activity in the medial prefrontal cortex (mPFC; [77–79]) and increased activity in the temporoparietal junction (TPJ; [80]) during social tasks. Findings from adult (e.g., [81–84]) and lesion studies (mPFC, [85, 86]; TPJ, [87]) generally demonstrate a reliance on these areas to perform theory of mind tasks. Likewise, in their longitudinal analysis of 7- to 30-year-olds, Mills et al. [88] found gray matter volume and cortical thickness in the mPFC and TPJ to decrease from childhood into adulthood. Similarly, the surface area for these regions peaked in early/pre-adolescence (i.e., 8- to 11-years-old) before decreasing

during adulthood. Therefore, the functional changes and anatomical development of these areas might reflect adaptations into how children understand the mental states of others.

Research into middle childhood is of particular interest, as it represents a period during which socioemotional abilities are greatly influenced by developments in self-awareness and adaptations to new environmental pressures [89]. In a longitudinal study measuring the development of cooperative, helping, and sharing behaviors in 6- to 12-year-olds, Malti et al. [90] found each of these prosocial behaviors to follow unique trajectories. Cooperation improved over time; however, rather than identifying an anticipated improvement in helping behavior, children actually decreased in their willingness to do so. Likewise, sharing followed a nonlinear inverted U-shaped trend, with peak sharing occurring at around 9 years of age.

### Socioemotional development and music

In the music cognition literature, the exploration of socioemotional skills is obscured by inconsistent operationalizations of core constructs (e.g., empathy, [91]; as cited by [92]), differences on music education approaches and contextual factors such as one's culture. For instance, joint music making has been found to both increase [27] and have no effect [63] on sharing behaviors. A similarly nuanced phenomena was found among preschoolers, whereby engaging in musical play increased spontaneous helping and overall sharing, but not spontaneous sharing [93]. Importantly, Rabinowitch, Cross, and Burnard [94] also found that engagement in a 9-month music program, focused on playing musical games specialized in encouraging empathy-inducing behaviors (e.g., entrainment, imitation, shared intentionality), improved trait empathy in school-aged children.

Although unmeasured in the present study, a large body of work has also investigated the relationship between self-esteem and music instruction. In a study by Rickard et al. [95], school-aged children received either Kodály music classes or group string instrument training in a standard classroom environment. They found that music classes helped combat a decline in self-esteem experienced by those in the control condition. Conversely, Costa-Giomi [96] randomly assigned children into two groups–one which received individual piano lessons and the other no formal music instruction. The self-esteem of children in the experimental condition increased significantly over three years; however, at no point did their scores differ from the control group. Finally, in a large-scale investigation of school-based musical instruction, Rickard, Bambrick, and Gill [97] found no significant improvements on a comprehensive battery of psychosocial well-being measures assessing self-esteem, attitude towards school, social skills, depression, etc.

Importantly, research that fails to identify a relationship between formal music instruction and socioemotional development often consist of one-on-one music instruction (e.g., [96, 98–100]). By contrast, research on the involvement of group-based musical activities generally supports a positive association with socioemotional development or some of its components (e.g., [62, 94, 101–104]), although research revealing few differences between music students and other groups (e.g., dancers) also exist [105]. Finally, it is also important to account for the fact that research on formal music instruction tends to be based on cross-sectional or short-term longitudinal designs, which may fail to capture the developmental changes of a long-term commitment such as music.

Using data collected from a 7-year longitudinal study, we aimed to understand the development of pitch-matching, rhythmic entrainment, and socioemotional skills in children who received formal music instruction and other non-music based after-school programs. However, due to disruptions caused by the COVID-19 pandemic, data collection was interrupted during the fifth year and was resumed the following year. Nevertheless, the lasting impact of

the pandemic caused many participants to drop out of the study, yielding a smaller sample size from year 4 onwards. As such, the present analysis only includes data from baseline to year 4 for a total of 5 time points. In addition, we aimed to explore the social implications of pitch-matching and replicate the positive relationship between rhythmic entrainment and socioemotional skills. Informed by these aims, we evaluated the following research questions:

1. What is the developmental trajectory of pitch-matching, rhythmic entrainment, and socioemotional skills for children enrolled in music or sports after-school programs compared to those who did not?

2. To what extent does pitch-matching and rhythmic entrainment predict performance on sharing behavior, trait empathy, theory of mind, and emotion-recognition?

## Methods

Data and code for analysis are available at https://osf.io/y6dfc/.

### Participants

Eighty-three children (ages 5.73–8.01 years old at baseline data collection, $M = 6.81$ years, $SD = 0.54$ years; 59% male, 41% female) were recruited from community and enrichment programs of the greater Los Angeles area to participate in a 7-year longitudinal study. All children self-selected into their program. Children were also raised in bilingual communities but attended English speaking schools. Research protocols were approved by the University of Southern California Institutional Review Board (UP-12-00322). Verbal assent was obtained from all child participants, and informed written consent was obtained from the parent or guardian in their preferred language. Children received toys and small prizes while parents received monetary compensation for their child's participation. Data collection occurred between September 17th, 2012, and March 4th, 2020. Subsequently, data was accessed on September 21st, 2022 and analyzed anonymously without access to the participants' identities.

Given the longitudinal nature of this study, not all participants could be evaluated with every measure at every time point (see S1 Table). To evaluate the relationship between gender and an incomplete set of observations, a Chi-Square Test of Independence was performed indicating that there was no relationship between gender and missing data ($\chi^2_1 < 0.001$, p = 1). Likewise, Wilcoxon Rank Sum Tests verified that individuals with missing data did not differ in age, cognitive ability, or any of the dependent variables analyzed in this study.

**After-school programs.** All after-school programs were free and intended for families from underserved communities in Los Angeles. Two cohorts of approximately age-matched participants were enrolled in either a music ($n = 26$) or sports ($n = 28$) program. A similarly matched control group ($n = 29$) was recruited from public schools in Los Angeles, consisting of children uninvolved with any systematic after-school program at the time of recruitment. We continued to monitor this group's participation in after-school programs through extensive yearly parental interviews to ensure they did not engage in any equally intensive, formal music instruction or sports program over the course of the study.

Participants in the music group were enrolled in the Youth Orchestra of Los Angeles at the Heart of Los Angeles. This intensive music program followed the Venezuelan "El Sistema" approach towards music education, which emphasizes a communal approach to music making via sectional, ensemble, and orchestral practices. This approach to music education was created to aid children and youths from underserved and at-risk communities socialize with one another in an inclusive and collaborative musical setting [see 106–108 for more]. Children in

this program met four to five times a week for 6 to 7 hours, during which, they learned about the fundamentals of music, engaged in group lessons on the violin or viola, and occasionally performed recitals. Additionally, students received some choir or group singing training.

Participants in the sports group were enrolled in either a community-based swimming ($n$ = 13) or soccer ($n$ = 15) program. Beyond the development of sport-related skills, both programs taught children the importance of health, nutrition, and sportsmanship. As with formal music instruction, sports are demanding sensorimotor activities that require long-term attentive practice to develop. Importantly, participants in the sports group did not engage in any systematic or long-term formal music instruction as verified by systematic annual interviews with their families. Children in the swimming program attended two one-hour training sessions each week and attended competitions on the weekend. Those in the soccer program attended three two-hour sessions every week, in addition to a one-hour game on the weekend.

### Research instruments

**Pitch-matching task.**   Pitch-matching was measured through a singing task, during which participants repeated back 10 melodies sung by an experimenter (Fig 1). Melodies were sung by vocally trained experimenters using the neutral syllable "la" and prompted the participant to respond with "la" in return [see 109 for more]. Each melody was 5–9 notes long in either the major or minor mode. Participant and experimenter audio were recorded as either an MP3 or WAV file, and then converted to the frequency-domain using a Discrete Fourier Transform available in Melodyne. In cases where participant responses were affected by vocal fry, the calculated Hertz value would be an octave lower than audibly perceived. These values were corrected by doubling the frequency of the pitch as to raise the pitch by one octave.

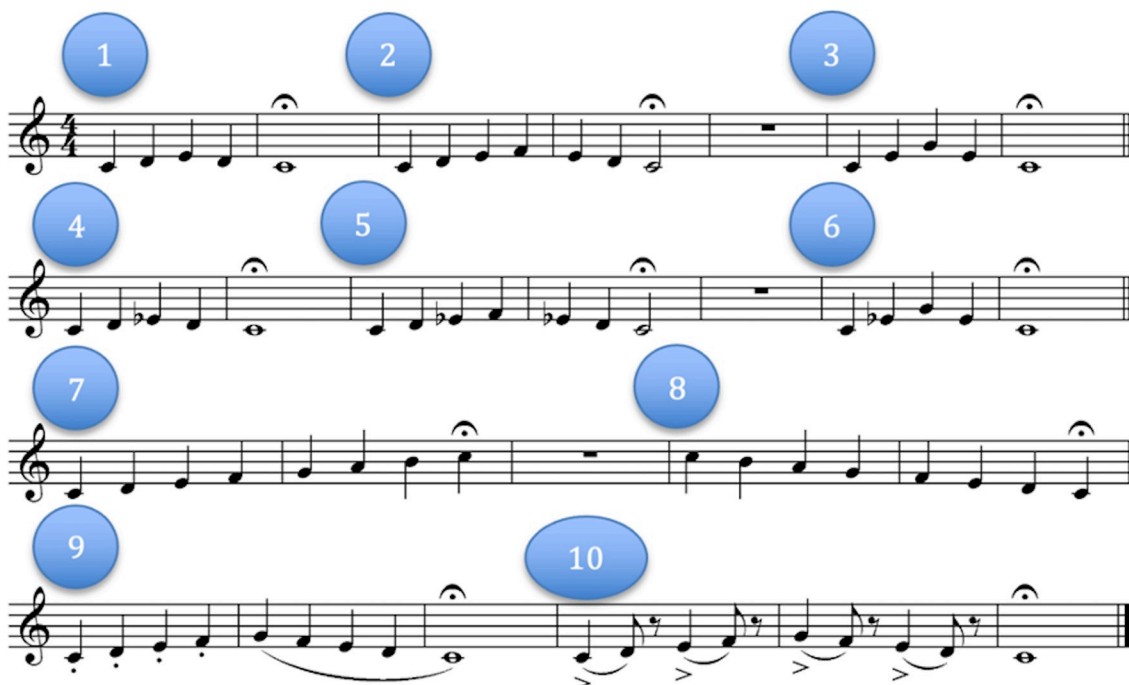

**Fig 1. Pitch-matching task stimuli.** Musical score showing the melodies sung during the task. Each numbered section represents a melody the participant first heard then echoed back to the experimenter. Sections 1–3, 4–6, 7–8, and 9–10 were grouped into four musical phrases during analysis.

Pitch-matching values were calculated in a similar manner to Pfordresher and Brown [110] by comparing the Hz of a target and response pitch. For our study, the target pitch was the audio sung by the experimenter, while the response pitch was that sung by the participant. For each of the 10 melodies, the difference between the experimenter and participant's pitches were calculated to represent an individual pitch deviation score. These scores were first averaged, then subtracted from 100, then divided by 100 to create a normalized singing score. In the case that combined individual pitch deviation scores were greater than 100Hz, indicating roughly twice the deviation from the target pitch, the sample was scored as a zero. The 10 normalized scores were then averaged to create an overall pitch-matching score ranging from 0 (speech/chant-like or a difference greater than two steps from the target pitch) to 1 (no difference in frequency between the sung pitch and the target pitch).

Participants at baseline were only asked to repeat sections 1–3 and 7–8, which came from component 6 of the AIRS Test Battery of Singing Skills [111]. Melodies 4–6 and 9–10 were developed by Ilari, Fesjian, and Habibi [112] to supplement those from year 1 onward.

**Rhythmic entrainment.** Rhythmic entrainment was measured following Kirschner and Tomasello [113]. Children were told to drum at different tempi with either a pre-recorded beat, or together with the pre-recorded beat and an experimenter [see 112]. These "alone" and "social" conditions were counterbalanced during presentation to prevent ordering effects. During the task, participants sat across from the experimenter with a drum placed in front of them. Stimuli would play from a laptop through a guitar amplifier to which participants would then drum along. A microphone inside the drum recorded the audio for each participant's response. This response audio was then compared to the audio of the original stimulus using a custom MATLAB [114] script applying circular statistics to measure their synchronization accuracy along a moving time window [see 98]. Values ranged from 0–1, with a value of 0 representing low synchronization accuracy and 1 representing perfect synchronization. Rhythmic stimuli included isochronous 120- and 180-bpm beats. The 120-bpm condition was administered each year; however, the 180-bpm was not administered in year 1 of the study.

**Sharing task.** From years 2 to 4 of the study, sharing was measured through a variation of the dictator game previously used and validated in an investigation of sharing in a population of 5-year-old children [115]. In year 2, children were asked to choose 10 stickers as rewards for completing different tasks throughout their behavioral assessment. Afterwards, they were given the option to share their stickers with a pictured stranger of the same age and gender. Participants were explicitly told that sharing would result in no benefit to themselves, and that no one would know how many stickers, if any, they shared. After receiving these instructions, children were left alone for 2–3 minutes to place however many stickers they wanted to share in a separate bag. Once complete, the number of stickers was counted by an experimenter while another researcher distracted them in a separate room. In years 3 and 4 of the study, the sharing task was modified to include 15 total stickers, as well as an additional option for children to share with a friend of similar age and no familial relation. This was done to make the sharing task more complex and age-appropriate; however, for the purposes of this analysis the total proportion of stickers given away was used as a measure of sharing to make the task comparable across time.

**Trait empathy, state empathy, and theory of mind.** From baseline to year 4, children completed the Index of Empathy for Children and Adolescents (IECA; [116]), the Reading the Mind in the Eyes Test of theory of mind (Eyes Test; [117]), and a modified variation of the Fiction Emotion-Matching task assessing state empathy [118]. The IECA is a 22-item questionnaire frequently used to assess the trait empathy of children and adolescents. Participants are prompted with phrases such as "People who kiss and hug in public are silly.", or "Some songs make me so sad I feel like crying.", to which they must respond yes or no. The Eyes Test is a

ubiquitous measure for theory of mind, during which participants view 28 black-and-white photos of an adult's eyes. Children are then asked to evaluate and select one of four options representing the emotional state of the pictured individual.

During the Fiction Emotion-Matching task, participants viewed a set of fictional movie clips exemplifying a target emotion (i.e., happiness, sadness, fear) and then evaluated the emotional state of the character. After completion of the clip, they were asked to explain what emotion the character was feeling, their own emotional state, and the degree to which they felt concerned for the character. Participants were scored based on the congruency of the character's identified emotion and their own emotion. Scores were either 0 (incongruent) or 1 (congruent) for each clip, which were then averaged together as a holistic evaluation of both cognitive and affective empathy. At baseline, participants viewed four 30-second clips from *Charlie & the Chocolate Factory* (see [118]). From years 1 through 4, more age-appropriate clips were selected from *The Land Before Time* (sad), *My Girl* (sad), *Jumanji* (fear), *Little Miss Sunshine* (happy), *American Girl* (sad), and *The Sandlot* (fear). Goldstein and Winner originally scored wholly exact matches as correct; however, we also opted to score conceptually similar emotions as correct if both emotions were generally aligned in terms of valence and arousal. For example, answering "excited" in response to the clip from *Little Miss Sunshine* (happy) would result in a correct match.

**Cognitive ability.** Several measures of general cognitive ability were also collected and included as covariates as they have been previously found to predict musical abilities [119, 120]. To measure general cognitive ability, each participant completed the Block Design, Matrix Reasoning, Vocabulary, and Similarities subtests of the Wechsler Abbreviated Scale of Intelligence (WASI-II; [121]). The Block Design and Matrix Reasoning tasks are well validated measures for problem solving and visuospatial skills. Likewise, the vocabulary and similarities subtests are measures of general knowledge or crystallized intelligence. In the Block Design task, participants saw and were asked to recreate a pictured design using a set of blocks. In the Matrix Reasoning task, participants were shown a group of images and asked to select which pattern, from a set of four or six options, completes the image. The Vocabulary Test had children orally define different words, and the Similarities Test had children orally describe the similarities between two common objects or concepts. A Full-Scale Intelligence Quotient (FSIQ-4) score was then calculated as a composite measure of all four subtests.

Participants also completed the Digit-Span task as a measure of auditory short-term memory (forward condition) and working memory (backwards condition). The experimenter read-aloud a sequence of numbers for the children to repeat back either in the order they heard it (forwards) or in reverse (backwards) order. As participants responded correctly, the number of digits in a sequence increased until they responded incorrectly to two trials in a row.

## Data collection procedure

Children were tested individually on a large battery of cognitive, socioemotional, and motor tasks each year over the course of seven years at either the program site or at the Brain and Creativity Institute at the University of Southern California, over several days. Testing could occur over multiple sessions depending on participant availability.

## Data analysis

Analysis was conducted in R [122] and processed using the tidyverse [123]. Linear mixed effects models were built using the lme4 package [124], with p-values generated by lmerTest [125]. For issues in convergence, the afex package [126] was used to identify optimizers. The

**Table 1. Correlation matrix of socioemotional measures.**

| Variables | Sharing | IECA | Eyes Test | Emotion-Matching |
|---|---|---|---|---|
| Sharing | — | | | |
| IECA | 0.23** | — | | |
| Eyes Test | 0.14 | 0.21*** | — | |
| Emotion-Matching | 0.06 | 0.24*** | 0.07 | — |

Matrix depicting the Pearson's correlation coefficients between each socioemotional measure. The Index of Empathy for Children and Adolescents is a measure of trait empathy, the Eyes Test a measure of theory of mind, and the Emotion-Matching task a measure of state empathy.

present analysis only includes data from baseline to year 4 due to the effects of the COVID-19 pandemic on data collection. An alpha value of $p < .05$ was used to determine significance for all analyses.

Correlations between measures of empathy and sharing were calculated as an estimation for construct validity within the socioemotional domain. Some but not all correlations were significant, with the few significant Pearson's $r$ ranging from 0.21 (Eyes Test vs. IECA) to 0.24 (Fiction Emotion-Matching vs. IECA), suggesting that each measure assessed a separate dimension of social cognition (see Table 1).

Principal component analysis (PCA) was used to reduce the dimensionality and multicollinearity of our regression analysis by creating a single composite score from measures intended to assess the same construct. The resulting outcome variables were the product of a PCA conducted on all the available data at every time point. This allowed us to best capture time-related changes and maximize comparability between years.

For the pitch-matching task, Pearson's $r$ ranged from 0.68 (phrases 2 and 3) and 0.88 (phrases 1 and 2). The resulting component representing "pitch-matching" was extracted from the scores for each musical phrase. Although the pitch-matching task was administered each year, the full task involving all four musical phrases was not administered until year 1 of the study. Values for pitch-matching at baseline were not generated for this year given the incomplete data. The subsequent component explained 82.5% of the variance, with factor loadings ranging from 0.84 to 0.94.

The component "rhythmic entrainment" was extracted from participant's synchronization accuracy to the 120- and 180-bpm stimuli in the alone and social conditions; however, data from the 180-bpm condition were not collected during Year 1. Consequently, as is the case with pitch-matching baseline, rhythmic entrainment values were not generated for year 1. Unlike the pitch-matching task, the range of correlations between the measures were markedly lower (120-bpm alone vs. 120-bpm social, $r = 0.36$; 180-bpm alone vs. 180-bpm social, $r = 0.68$). The resulting principal component explained 64% of the variance, with factor loadings ranging from 0.72 to 0.88.

A PCA was not conducted on the socioemotional or cognitive measures due to their weak intercorrelations, in addition to better maintaining interpretability for each outcome variable.

## Growth curves

Linear mixed effects models were created to measure the growth of rhythmic entrainment, pitch-matching, sharing, and empathy over time, and to determine if any of these variables varied as a function of formal music instruction. Year, group (e.g., music, sports, and control), and their interaction were included as fixed effects in all models to evaluate these questions. FSIQ-4 and working memory were tested as covariates in all four models in line with previous literature [119]. In addition, auditory short-term memory was included as a covariate for the

pitch-matching and rhythmic entrainment models. Gender was included as a covariate for the socioemotional models since females have been shown to perform better at empathy-related tasks [116, 127]; however, later analysis would reveal that FSIQ-4 and working memory were the only preferred covariates. Model selection was performed by choosing the model with the lowest Bayesian information criteria (BIC) in order to penalize models with increased complexity. All models had a random effects structure including by-participant random intercepts and slopes for time to account for individual differences in rates of change. The music group was set as the reference level from which all comparisons were made.

## Results

### Music production

FSIQ-4 score was a significant predictor for pitch-matching, whereby every unit increase in FSIQ-4 improved pitch-matching by an average of 0.017 points ($\beta = 0.006$, $SE = 0.07$, $t = 2.603$, $p = .01$; $d = 0.20$, 95% CI [0.05, 0.35]; see S2 Table for model estimates). In addition, pitch-matching over time decreased significantly for individuals in the control condition relative to the music condition, with an average decrease in rate by 0.26 points ($\beta = -0.26$, $SE = 0.10$, $t = -2.51$, $p = 0.016$, $d = -0.36$, 95% CI [-0.65, -0.07]; see Fig 2). Year failed to reach significance ($\beta = 0.6$, $SE = 0.07$, $t = 0.80$, $p = 0.43$; $d = 0.24$, 95% CI [-0.03, 0.51], indicating no natural changes due to development alone over time. Since baseline PCA scores were not calculated, pitch-matching at this timepoint was evaluated by taking the average of the available data (phrases 1 and 3). This averaged value was then used as the outcome variable in a linear model predicted by group. Model estimates indicate that individuals in the music group scored about 0.13 points higher than those in the sports group ($\beta = 0.13$, $SE = 0.06$, $t = 2.15$, $p = 0.036$, $d = 0.65$, 95% CI [0.03, 1.26]; see Fig 3).

Rhythmic entrainment scores did not differ at baseline; however, there was a statistical trend for individuals in the control group to score 0.62 points less than the music group ($\beta = -0.62$, $SE = 0.32$, $t = -1.95$, $p = 0.056$; see Fig 3). Model estimates reveal that rhythmic entrainment is significantly predicted by year ($\beta = 0.29$, $SE = 0.04$, $t = 6.84$, $p < .001$; $d = 1.11$, 95% CI [0.87, 1.34]; see Fig 4), indicating a strong natural development in the ability over time. Although the model did not reveal any group differences in rates of change, when year is held constant, the control group scored about 0.70 points lower ($\beta = -0.70$, $SE = 0.29$, $t = -2.54$, $p = .013$, $d = -0.60$, 95% CI [-1.07, -0.12]) than the music group on the rhythmic entrainment task. Similarly, the sports group exhibited a statistical trend of scoring about 0.51 points less ($\beta = -0.51$, $SE = 0.29$, $t = -1.79$, $p = 0.08$, $d = -0.42$, 95% CI [-0.88, 0.05]) than the music group.

**Socioemotional skills.** A significant effect of time on sharing was found, with an average increase of 7.2% stickers given away each year ($\beta = 0.072$, $SE = 0.031$, $t = -2.28$, $p = .024$; $d = 0.53$, 95% CI [0.36, 0.70]). A significant group by year interaction was also found, whereby individuals in the sports conditions gave away stickers about 9% more than the music group each year ($\beta = 0.09$, $SE = 0.04$, $t = 2.054$, $p = .042$, $d = 0.17$, 95% CI [0.01, 0.33]). Additional analyses found the overall shape of the sharing distributions to follow an inverted U-shape trend over time (see Fig 5), peaking at year 3 ($M = 0.62$, $SD = 0.16$).

Measures of empathy did not show a uniform pattern of results with respects to group, year, or cognitive abilities, highlighting their lack of convergent validity (see Fig 6; see S3 Table for model estimates). For instance, the IECA had a main effect of working memory ($\beta = 0.018$, $SE = 0.006$, $t = 2.91$, $p = .004$; $d = 0.17$, 95% CI [0.05, 0.28]) while the Eyes Test was significantly predicted by time ($\beta = 0.05$, $SE = 0.008$, $t = 5.94$, $p < .001$; $d = 1.37$, 95% CI [1.05, 1.68]). However, the Fiction Emotion-Matching task found time ($\beta = 0.05$, $SE = 0.02$, $t = 3.04$, $p = .003$; $d = 0.16$, 95% CI [0.04, 0.28]) and the sport by year interaction to be significant. Specifically,

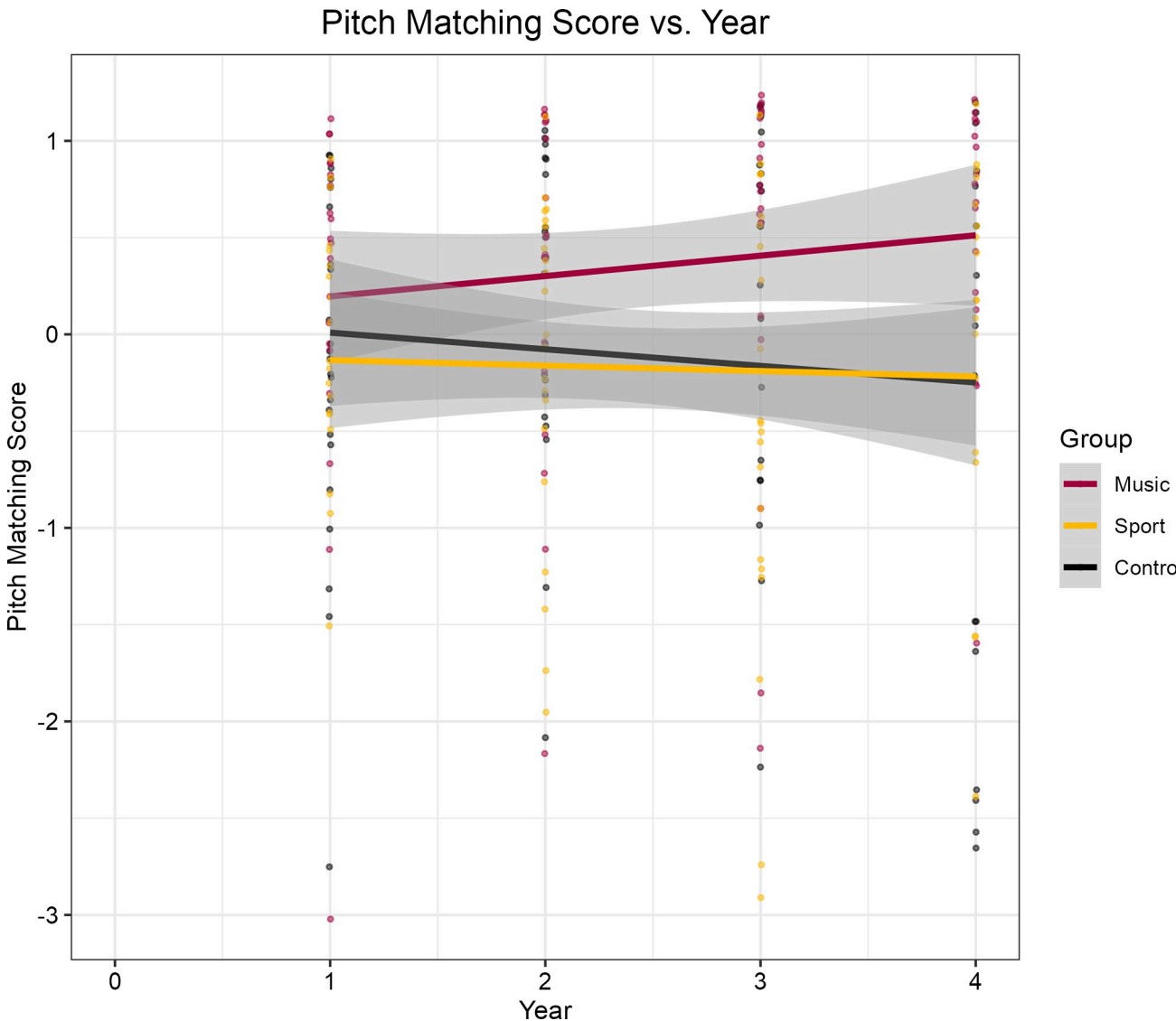

**Fig 2. Pitch-matching growth curve.** Pitch-matching score plotted against time. Only individuals in the music group improved on pitch-matching ability over time. Grey shading indicates the 95% confidence interval for the lines of best fit.

the sports group showed a 0.05 points decrease relative to the music group ($\beta$ = -0.05, SE = 0.02, t = -2.32, p = .021, $d$ = -0.14, 95% CI [-0.26, -0.02]), indicating that the music group's performance improved more over time on this task.

## Music production and socioemotional skills

A second set of analyses was conducted predicting socioemotional skills as a function of music production (i.e., pitch-matching or rhythmic entrainment). Year and by-participant random slopes for time were again included in all models as fixed and random effects, respectively. FSIQ-4 and working memory were added as covariates and evaluated by their model's respective BIC.

As shown in Table 2, each unit increase in pitch-matching ability decreased the proportion of stickers shared by about 3.5% ($\beta$ = -0.035, SE = 0.01, t = -2.38, p = .019; $d$ = 0.19, 95% CI

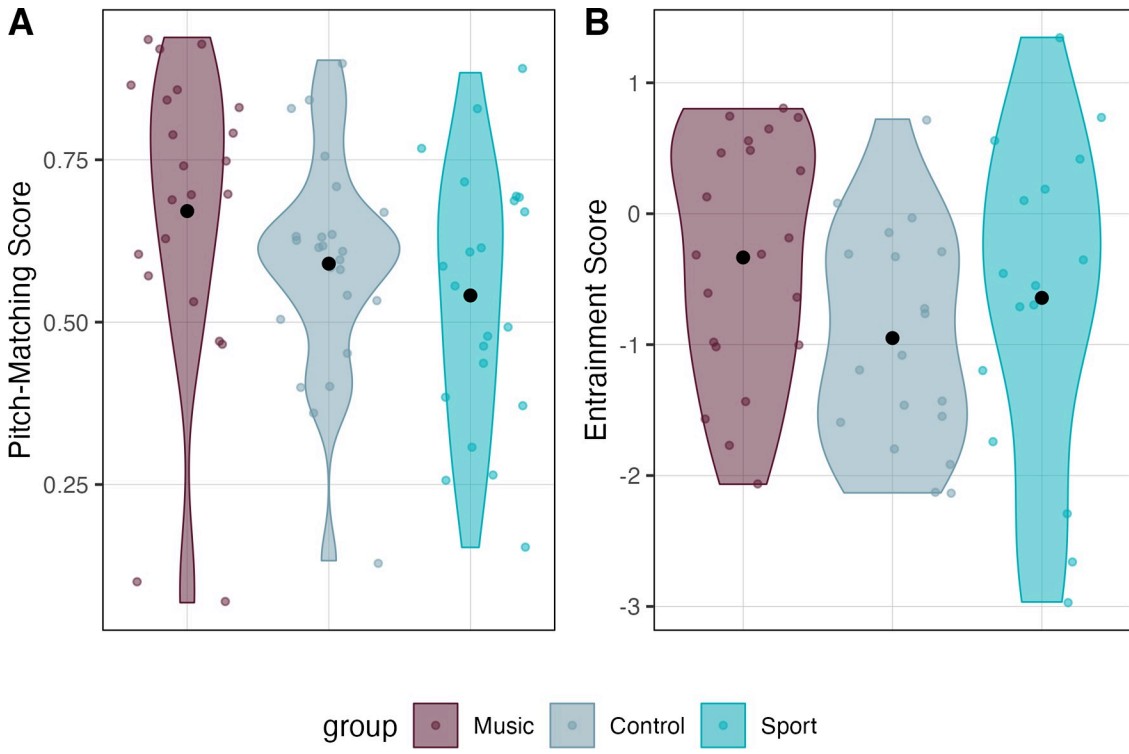

**Fig 3. Baseline pitch-matching and rhythmic entrainment scores.** Violin plot depicting pitch-matching and rhythmic entrainment scores at baseline. (A) Pitch-matching scores were calculated as the average pitch-matching score of phrases 1 and 3 (Music, $M = 0.67$ $SD = 0.24$; Control, $M = 0.59$, $SD = 0.17$; Sport, $M = 0.54$, $SD = 0.20$). (B) Baseline rhythmic entrainment scores were generated by conducting a PCA on all available rhythmic entrainment data (Music, $M = -0.33$ $SD = 0.91$; Control, $M = -0.95$, $SD = 0.83$; Sport, $M = -0.64$, $SD = 1.25$). Colored dots visualize individual observations with black dots indicating group averages.

[0.03, 0.34]). As with the analysis of pitch-matching's growth curve, time continues to be a robust and significant predictor in the model. On the other hand, rhythmic entrainment failed to reach significance as a predictor in the sharing model.

Few consistent patterns were found between the music production measures and empathy; one of which being the failure for pitch-matching to predict any empathy measure (Table 3). Rhythmic entrainment displayed a trend level relationship in both the IECA (p = 0.07, d = 0.16, 95% CI [-0.02, 0.34]) and the Eyes Tests (p = 0.08, d = 0.12, 95% CI [-0.01, 0.26]), but failed to reach significance in the emotion-matching model (see Table 4).

Instead of finding an effect of music production on socioemotional outcomes, an effect of time was found for most models. Interestingly, the effect of year on IECA and Fiction Emotion-Matching performance differed depending on which music production measure was included as a fixed effect; IECA score was significantly predicted by year (*p* = 0.016, *d* = 0.34, 95% CI [0.08, 0.60]) only when pitch-matching was included in the model. Likewise, Fiction Emotion-Matching score was only significant when rhythmic entrainment was included in the model (*p* = 0.024, *d* = 0.15, 95% CI [-0.04, 0.25]).

The only empathy measure to consistently improve over time was the Eyes Test, which increased each year by an average of 0.035 points (*β* = 0.035, *SE* = 0.006, *t* = 6.26, *p* < .001; *d* = 0.81, 95% CI [0.52, 1.10]) or 0.052 points (*β* = 0.052, *SE* = 0.006, *t* = 9.01, *p* < .001; *d* = 0.92, 95% CI [0.68, 1.16]) in the pitch-matching and rhythmic entrainment models, respectively. Notably, the base model predicting task performance as a function of pitch-matching initially failed to converge and required the bobyqa optimizer to reconcile this issue. Importantly, the

## Rhythmic Entrainment Score vs. Year

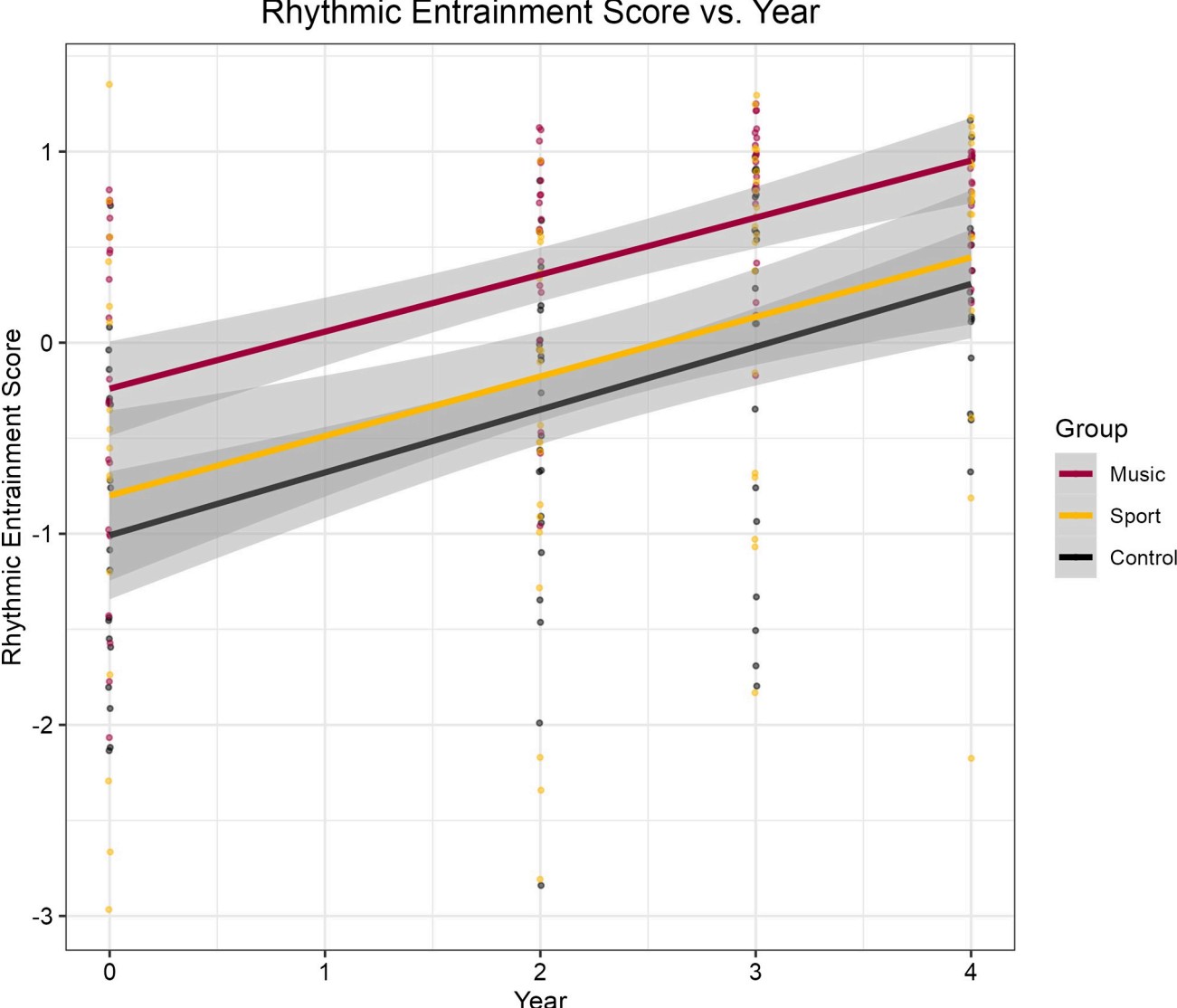

**Fig 4. Rhythmic entrainment growth curve.** Rhythmic entrainment score plotted against time. There is a strong linear trend for all groups in relation to year; however, no significant differences in the rate of change between groups. Grey shading indicates the 95% confidence interval for the lines of best fit.

effect sizes for all inconsistent interpretations for time were trivial or small, ranging from a Cohen's d of 0.24 (Fiction Emotion-Matching performance vs Year), to d = 0.31 (IECA score vs. Year). On the other hand, the effect sizes for time in the Eyes Test models were much larger at a Cohen's *d* of 0.81 to 0.92 and exhibited a shared finding across models given the effect's robustness.

## Discussion

### Pitch-matching and rhythmic entrainment development

The present longitudinal study aimed to explore the development of pitch-matching, rhythmic entrainment, and socioemotional skills in a population of 6-year-old children enrolled in either a music, sports, or no after-school program. We also aimed to identify associations between our musical and socioemotional measures. When asked to mimic a set of sung

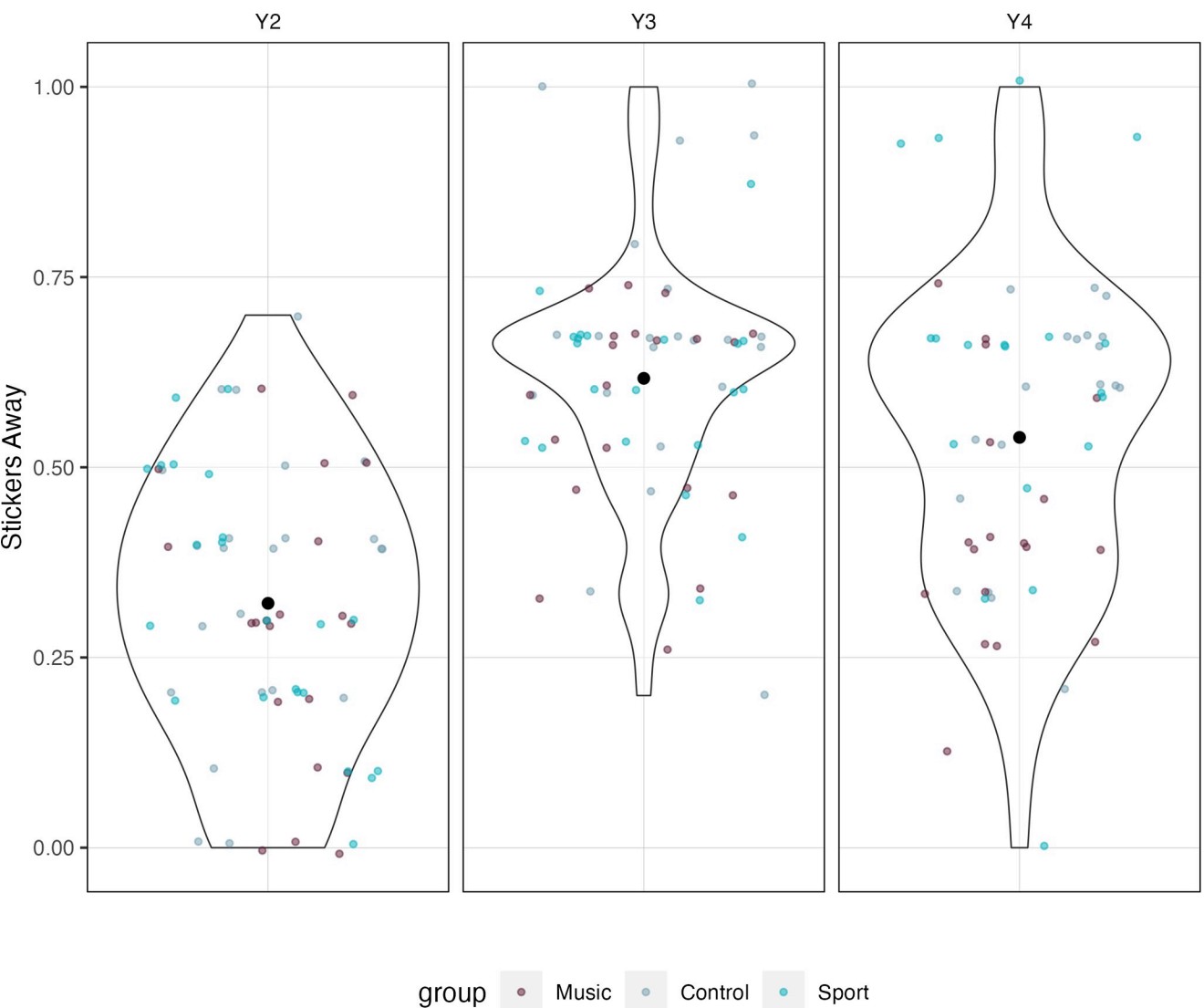

**Fig 5. Distribution of stickers given away.** Violin plot depicting the proportion of stickers given away to either a stranger or a friend. The proportion of stickers shared followed an inverted U-shaped trend, with an average of 32% of stickers ($SD = 0.18$) given away at year 2, 62% ($SD = 0.16$) at year 3, and 54% ($SD = 0.20$) at year 4. Colored dots represent individual participants and their respective group.

pitches, individuals from the music group generally outperformed their sport and control counterparts. It comes as no surprise that over the course of four years, formal music instruction significantly improved pitch-matching relative to controls; however, there were also no significant differences between the music and sport groups.

Interestingly, there were also no time-related improvements in pitch-matching for the control and sports groups. Pitch-matching has been shown to improve naturally over time in both cross-sectional [128–131] and longitudinal studies [132], and even in populations lacking formal music instruction [133]. Yet in most of these studies, participants were asked to complete shorter tasks, often involving 2–3 pitches instead of longer pitch melodies and scales, as in the present study. Child musicians in our study were primarily violin and viola players, which are melodic instruments. Violinists and violists are used to playing short melodies and practicing scales. We suggest that the improvements of child musicians on pitch-matching over time is an indication of near transfer.

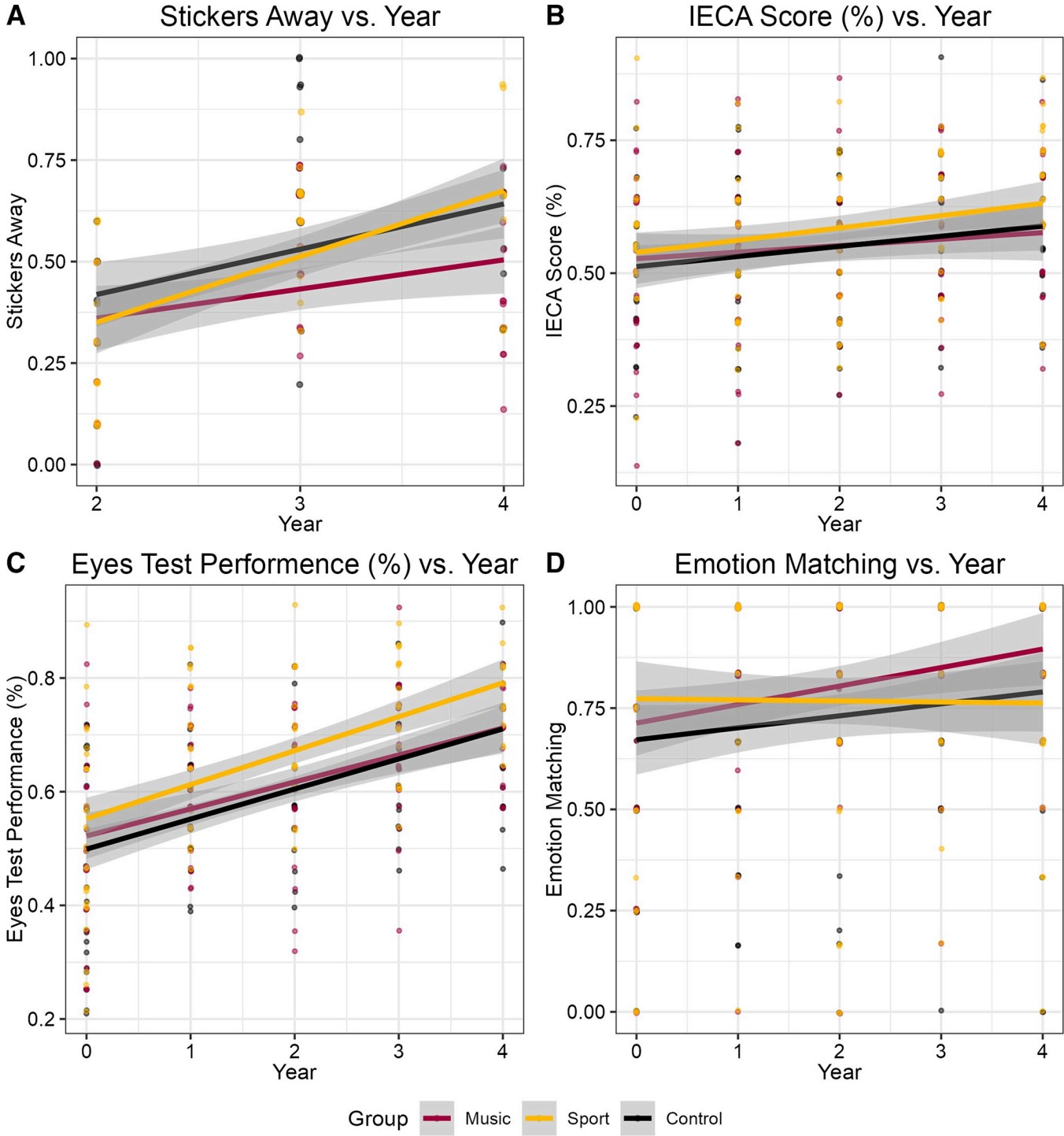

**Fig 6. Socioemotional skills growth curves.** Measures of socioemotional skills plotted against time. Performance on the (A) sharing, (B) IECA and (C) Eyes Test improved over time, while performance on the (D) Fiction Emotion-Matching task did not. No differences in rate of change were found in any measure apart from the Emotion-Matching task, where individuals in the sports group performed on average 0.05 points worse than the music group each year ($p = 0.021$). Grey shading indicates the 95% confidence interval for the lines of best fit.

**Table 2. Music production scores predicting sharing with a stranger.**

| Effect | β | SE | t | p | 95% CI |
|---|---|---|---|---|---|
| Sharing ~ pitch-matching | | | | | |
| Intercept | 0.121 | 0.058 | 2.119 | 0.036 | [0.01, 0.24] |
| pitch-matching | -0.035 | 0.015 | -2.379 | 0.019 | [-0.06, -0.01] |
| Year | 0.128 | 0.019 | 6.752 | < 0.001 | [0.09, 0.17] |
| Sharing ~ rhythmic entrainment | | | | | |
| Intercept | 0.140 | 0.059 | 2.368 | 0.020 | [0.02, 0.26] |
| rhythmic entrainment | -0.002 | 0.019 | -0.148 | 0.882 | [-0.04, 0.03] |
| Year | 0.120 | 0.020 | 5.922 | < 0.001 | [0.08, 0.16] |

CI = confidence interval; Regression model estimates predicting proportion of stickers given away as a function of either pitch-matching or rhythmic entrainment ability. Sharing was found to vary negatively with respects to pitch-matching, while no significant relationship with rhythmic entrainment was found. In general, pitch-matching ability reduced the proportion of stickers given away by about 3.5%. Year was again a significant predictor in both models.

Analysis also revealed cognitive ability to have a significant yet small relationship with pitch-matching. As evidenced by studies on the far-transfer effects of formal music instruction on visuospatial (e.g., [134–136]) and verbal memory (e.g., [137–142]), musical ability appears to robustly associate with general cognition. For instance, Berkowska and Dalla Bella [137] found that reducing the linguistic complexity of a target stimulus decreased cognitive load, subsequently improving pitch-matching performance. In line with their findings, individuals with higher FSIQ-4 scores in our study were likely better equipped to store pitch information and later reproduce them.

In contrast to pitch-matching, where time-related improvements were exclusive among musicians, participants from all conditions significantly improved in their ability to rhythmically entrain each year. Nevertheless, no musician's advantage was found; the rate of improvement did not change as a function of group. We speculate that this finding could be related to

**Table 3. Pitch-matching predicting empathy measures.**

| Effect | β | SE | t | p | 95% CI |
|---|---|---|---|---|---|
| IECA ~ pitch-matching | | | | | |
| Intercept | 0.425 | 0.035 | 12.423 | < 0.001 | [0.37, 0.50] |
| pitch-matching | -0.002 | 0.010 | -0.161 | 0.873 | [-0.02, 0.02] |
| Year | 0.021 | 0.008 | 2.485 | 0.016 | [0.00, 0.04] |
| working memory | 0.018 | 0.007 | 2.542 | 0.012 | [0.00, 0.03] |
| Eyes Test ~ pitch-matching | | | | | |
| Intercept | 0.576 | 0.018 | 32.811 | < 0.001 | [0.54, 0.61] |
| pitch-matching | 0.002 | 0.008 | 0.245 | 0.806 | [-0.01, 0.02] |
| Year | 0.035 | 0.006 | 6.258 | < 0.001 | [0.02, 0.05] |
| Fiction Emotion-Matching ~ pitch-matching | | | | | |
| Intercept | 0.772 | 0.043 | 18.162 | < 0.001 | [0.69, 0.86] |
| pitch-matching | 0.004 | 0.017 | 0.218 | 0.827 | [-0.03, 0.04] |
| Year | 0.014 | 0.013 | 1.085 | 0.280 | [-0.01, 0.04] |

CI = confidence interval; Regression model estimates predicting empathy measures as a function of pitch-matching. Pitch-matching failed to predict any empathy measures at both a significant and trending level. The driving predictor for most tasks was instead time, which improved performance on IECA by about 0.02 points (p = 0.016) and the Eyes Test by about 0.03 points (p < .001) each year. For each unit increase in working memory, IECA score also improved by about 0.02 points (p = 0.012). The Fiction Emotion-Matching task failed to vary as a function of either predictor.

**Table 4. Rhythmic entrainment predicting empathy measures.**

| Effect | β | SE | t | p | 95% CI |
|---|---|---|---|---|---|
| IECA ~ rhythmic entrainment | | | | | |
| Intercept | 0.537 | 0.018 | 29.477 | < 0.001 | [0.50, 0.57] |
| rhythmic entrainment | 0.019 | 0.011 | 1.805 | 0.074 | [-0.00, 0.04] |
| Year | 0.013 | 0.007 | 1.726 | 0.088 | [-0.00, 0.03] |
| Eyes Test ~ rhythmic entrainment | | | | | |
| Intercept | 0.517 | 0.019 | 27.001 | < 0.001 | [0.48, 0.56] |
| rhythmic entrainment | 0.018 | 0.010 | 1.764 | 0.079 | [-0.00, 0.04] |
| Year | 0.052 | 0.006 | 9.006 | < 0.001 | [0.04, 0.06] |
| Fiction Emotion-Matching ~ rhythmic entrainment | | | | | |
| Intercept | 0.684 | 0.043 | 16.022 | < 0.001 | [0.60, 0.77] |
| rhythmic entrainment | 0.035 | 0.025 | 1.431 | 0.154 | [-0.01, 0.08] |
| Year | 0.032 | 0.014 | 2.298 | 0.024 | [0.00, 0.06] |

CI = confidence interval; Regression model estimates predicting empathy as a function of rhythmic entrainment. Rhythmic entrainment failed to reach significance for all empathy measures; however, was trending towards significance for both the IECA ($p = 0.07$) and Eyes Tests ($p = 0.07$). Year was the only significant predictor in any of the models, with an average increase of about 0.05 points in the Eyes Test ($p < .001$) and 0.03 points on the Fiction Emotion-Matching task ($p = 0.024$) over time. Year was trending towards significance ($p = 0.09$) in the IECA model.

the nature of music learning in an orchestral setting, particularly among beginning string players. It would be interesting for future studies to study the development of rhythmic entrainment skills within orchestral musicians, as there could be differences between instruments that typically play melodic materials (e.g., violins, flutes) and those that help sustain the underlying beat of the ensemble (e.g., percussion). Previous studies on the rhythmic entrainment skills of expert adult instrumentalists have produced contrasting findings (e.g., [143, 144]), and research on developing musicians may offer additional insights.

A common limitation in longitudinal studies is also the fragmented nature of data collection. As studies continue, existing measurements are updated to be more age appropriate, participants might be unavailable to test one year due to personal circumstances, and others may lose interest and dropout from the study entirely. The present study is no stranger to these issues, resulting in incomplete baseline pitch-matching and year 1 rhythmic entrainment data. It is possible that the specific phrases of pitch-matching unmeasured at baseline (phrases 2 and 4) are fundamentally more challenging for 6-year-old children to remember or produce given their increased note length [e.g., 145–147], and that natural improvements between the ages of 6 and 7 are substantial enough to alter the overall trajectory of the model. This assumption is not unreasonable as phrases 2 and 4 were designed to incorporate musical elements the AIRS Test Battery of Singing Skills (phrases 1 and 3) lacked, such as the minor mode, articulation, and expressivity. Likewise, it could be argued that rhythmic entrainment to a 180-bpm stimulus is a challenging task, particularly for child nonmusicians, as it strays far away from a child's spontaneous motor tempo [4, 19]. However, due to the absence of year 1 data, we are unable to determine the training-related differences associated with rhythmic entrainment at this tempo.

Of note, our analysis suggests that there may have been some baseline differences in the measures of pitch-matching and rhythmic entrainment. Specifically, at baseline, participants in the music program outperformed those in the sports group on pitch-matching while a group difference between the music and control groups approached significance in the rhythmic entrainment task. Combined with the fact that children self-enrolled into their programs,

there is a chance that these baseline differences are influenced by selection bias. Parents of children who enrolled in the music program may have home environments that valued musical exposure and more frequently participated in musical activities (e.g., singing/dancing at home, attending musical events with family, etc.). This early exposure to music might have been particularly beneficial for the child's early musical development. Additionally, children with genetic predispositions or preexisting cognitive advantages may be more inclined to enroll and remain in the music program, complicating our ability to determine the driving factor for any changes in our outcome variables. As such, our analysis remains correlational in nature.

We note that Ilari, Fesjian, and Habibi [112] from our group analyzed the baseline rhythmic entrainment data to the 120-bpm stimuli for the same dataset and found no significant group differences in the alone condition; however, they did find a significant difference between the music and control groups in the social condition. Similarly, Wood, Perez, et al. [148] performed a PCA on the same pitch-matching and rhythmic entrainment data and report no baseline differences in the resulting component they refer to as "music training". We believe these differences stem from the difference in the outcome variables and analysis models. The dataset analyzed by Ilari et al. [112] is most conservative in their methods where they used untransformed raw data and omitted all participants with missing data. In the report by Wood, Perez et al. [148] they extracted a single construct of music training skills combining all available data on rhythmic entrainment and pitch-matching. The present study acts as an intermediary by analyzing a transformed variable representing overall performance for rhythmic entrainment and pitch-matching separately. Similar to the findings by Ilari et al., which used raw untransformed data for their analysis of baseline performance, we show no differences in the alone condition for the 120-bpm stimuli, while a group difference between the music and control group approached significance in the social condition. For the 180-bpm stimuli, musicians significantly outperformed the control group in both the alone and social conditions. Our results confirm that these group differences continue to hold over-time. As for pitch-matching using the raw untransformed data, there were no group differences for phrase 1 of the pitch-matching task, but that musicians did outperform the sports group for phrase 3. Despite this baseline difference, the music group consistently improved overtime in their pitch-matching ability whereas the other groups did not.

## Socioemotional development

Children's sharing behavior was found to improve over time. As explained by Malti et al. [90], sharing is a costly behavior that rarely benefits the sharer. In one-off incidents like a laboratory experiment, the incentive to share is often reduced as the potential rewards associated with continued interaction are unavailable (e.g., building rapport with the recipient, strengthening a relationship, etc.). Children in our study replicated the results from Malti et al. by exhibiting an increase in sharing from ages eight to nine, followed by a decrease in sharing from ages nine to 10. As observed by Benenson, Pascoe, and Radmore [149] it is not uncommon for this transitional period between early and middle childhood to coincide with an increase in sharing behavior. However, with continued growth in perspective-taking and additional opportunities for peer interaction, children might instead become more mindful and selective with whom they share with [150].

A drawback to our design is that the prosocial task did not immediately follow rhythmic entrainment. At each time point, the sharing tasks were the last to be administered; the rhythmic entrainment and sharing tasks could be administered hours apart. Although some research has identified a trait effect of rhythmic entrainment on socioemotional skills [151], many of the previous studies on this topic have identified a more transient state effect of

rhythmic entrainment on socioemotional outcomes (e.g., [28, 60, 152]). It could be that rhythmic entrainment induces a state effect, whereby musicians are more prosocial during and immediately following rhythmic engagement. However, once they return to their everyday life, these prosocial gains revert to normal levels. Moreover, the rhythmic entrainment and sharing tasks were conducted between different pairs of individuals; the rhythmic entrainment task was between the participant and the experiment while the sharing task was between the participant and the representation (i.e., a paper bag) of an absent stranger or friend. The dynamics of this interaction may have changed if the receiver was instead the experimenter, or if the receiver was physically present.

Concerning the empathy domain, the growth curves for each measure followed separate and unique patterns. Bryant's IECA did not improve over time but did as a function of working memory. Conversely, the Eyes Test improved drastically over time without an effect of cognitive ability. Finally, the Fiction Emotion-Matching task was not moderated by any measure of cognitive ability but did include a greater yearly improvement for musicians relative to athletes. When interpreting these results, it is important to consider the intended purpose of each measure; Bryant's IECA broadly measures trait empathy (however, see [153]), the Eyes Test measures theory of mind (however, see [154, 155]), and the Fiction Emotion-Matching task is a holistic measure assessing multiple aspects of empathy (i.e., situational, cognitive, and theory of mind).

With regards to the IECA, Rabinowitch, Cross, and Burnard [94] found performance on this task to improve due to formal music instruction. However, it is important to note that their music intervention explicitly specialized in training empathic behaviors. Outside of this study, there is limited evidence to support the notion that musicians are characteristically more empathic than the general population, and this is reiterated in our results. Our findings are more similar to those of Cespedes-Guevara and Dibben [105], who did not find improvements on measures of trait or situational empathy. Both our study and Cespedes-Guevara and Dibben focused on large ensemble programs, and more specifically, music learning in orchestras. Although orchestral training programs have undergone some changes, they still tend to be highly structured and hierarchical. It is possible that music programs predicated on collaboration such as popular music, jazz and mariachi foster heightened empathy in developing musicians. This would be consistent with studies that found differences in empathy scores favoring popular musicians over classically-trained musicians [156].

As for the Eyes Test, this measure fundamentally requires participants to look into another's eyes and evaluate their emotional state. Very rarely do musicians probe for emotional information from someone's eyes alone; rather, eye-contact in a musical setting seems to be used primarily for coordinating technical aspects of the performance (e.g., entries and synchronizing rhythmic sections). In fact, studies tend to find null or mixed results when associating formal music training with the recognition of emotion from visual [157] and audiovisual [52, 158, 159] stimuli. In a longitudinal study conducted by Good et al. [52], small-n groups of children with cochlear implants were enrolled into either a music or visual-arts program. After 6-months of training, pre/post measures of auditory and audiovisual emotion-recognition were compared yielding no significant differences between groups. Importantly, much of this research comes from cross-sectional investigations of cochlear implant users, so additional longitudinal research with larger populations of individuals with normal hearing are required before relationships can be established.

Finally, success in the Fiction Emotion-Matching task requires participants to identify the emotion of a fictional character then emulate that same emotion. This task is unique among our empathy measures as it incorporates an auditory dimension, of which musicians are known to process differently than nonmusicians. In the context of our Fiction Emotion-

Matching task, musicians may be uniquely advantaged in utilizing their enhanced auditory processing to identify emotions conveyed by a movie soundtrack or a speaker's words. Although our study did not specifically assess auditory emotion recognition, we speculate that similar mechanisms may be at play—a musician's heightened sensitivity to emotional cues in auditory stimuli may lead to an improvement in identifying and ultimately embodying the emotions conveyed in a movie clip.

Additional covariates such as a general cognitive ability and working memory were also expected to moderate our outcome variables, as previous studies have demonstrated a strong, positive relationship between formal music instruction and cognitive ability [119, 120]. Schellenberg and Mankarious [160], for example, found formal music instruction to improve emotional comprehension; however, this effect disappeared when FSIQ scores were held constant. In the present study, effects of cognitive ability on model estimates were generally observed, but with a much weaker effect than anticipated. Similarly, although trending and even significant associations were found between socioemotional skills and music production abilities, each of these effect sizes were marginal to small in magnitude. Overall, time appears to be the most frequent and robust predictor for any improvements on socioemotional behaviors, which we come to expect over the natural course of human development and maturation.

The results of this study provide limited support for previous literature associating socioemotional abilities, formal music instruction, and their interaction (e.g., [94, 104, 112]). In line with conclusions from several meta-analyses the far transfer effects of music instruction in this study were either nonexistent [71, 72], or small and isolated to tasks with an auditory component [73], as seen with the emotion-matching task. Ilari et al. [112], which used the same data as this study, albeit at one individual time point, found a positive correlation between rhythmic entrainment ability and prosocial behavior in the music group. Although this correlation was strong, the relationship was only present between scores of rhythmic entrainment with a metronome (in the alone condition) and the number of stickers given to a friend, not a stranger. This study offers a more holistic assessment of the data by creating composite measures of rhythmic entrainment in all conditions, as well as scoring prosociality through the total proportion of stickers given away (e.g., [90]) instead of splitting it by the receiver. Although splitting the data by the receiver would allow for a more nuanced perspective into the sharing behavior of children, combining the conditions as we did here allow us to assess sharing behavior across a longer timeframe.

Our research also contributes to the growing body of literature identifying a relationship between formal music instruction and socioemotional gains when an auditory component is included in the stimuli. Much of the literature focuses on the identification of emotional states elicited by the face, except whole-body gestures remain a key aspect of musical engagement (e.g., bobbing and swaying, [161]). It would be interesting for future research to evaluate a musician's ability for emotion recognition from body language. This will help clarify whether their advantage is limited to the auditory domain. Moreover, while it is not uncommon to assess socioemotional skills in relation to formal music instruction, previous research tends to measure a comparatively short timeframe (e.g., 6-months, [27]; 9-months, [94]; 1-year, [105]). Yet, musical development is a lifelong process. Assessing behavioral outcomes from formal music instruction within a single year could be premature. Likewise, it is important to consider that any socioemotional outcomes derived from participation in a formal music program are dependent on the quality and objectives of the program. In that regard, participation in a non-formal music program such as compulsory classes offered at school might offer different results given the lack of structure and intensity; however, additional research is required. Our study focused on a music program based on the orchestral model, and placed emphasis on the development of musical skills through an intensive training process. While the program did

foster some social and collaborative skills, the primary goal of the program was not the development of socioemotional skills as in Rabinowitch et al. [94], nor to compare the experiences of musicians who practiced different genres including popular music [156, 162]. By measuring musical and socioemotional skills over the course of multiple years, we are uniquely positioned to capture the long-term growth resulting from formal music instruction and offer robust data on the complex construct that is musical development.

## Supporting information

**S1 Table. Sample sizes per year.** Sample sizes for each outcome measure. "x" indicates that no data is available for that year for that variable. Phrases two and four for the pitch-matching task and 180-bpm stimuli for the rhythmic entrainment task were not collected at baseline and year 1, respectively. As such, a PCA component was not generated for those years due to the incompleteness of the data.
(DOCX)

**S2 Table. Music production growth curve model estimates.** Regression model estimates predicting pitch-matching and rhythmic entrainment ability as a function of group, year, and their interaction. The pitch-matching model indicates that an increase in general cognitive ability improves pitch-matching ability by 0.20 points. Likewise, the change in pitch-matching for individuals in the control group decreased significantly over time relative to those in the music group by about 0.26 points ($p = 0.016$). The rhythmic entrainment model reveals that individuals in the music group scored about 0.70 higher than those in the control group ($p = 0.013$). Similarly, the music group trended towards outperforming the sports group by about 0.51 points ($p = 0.08$). Time was the strongest predictor improving rhythmic entrainment by nearly 0.29 points each year ($p < .001$).
(DOCX)

**S3 Table. Socioemotional skills growth curve model estimates.** Regression model estimates predicting socioemotional skills as a function of group, year, and their interaction. While sharing did not vary as a function of these predictors, measures of empathy did, albeit without consistency. The Index of Empathy for Children and Adolescents (IECA) improved by about 0.02 points (p = 0.004) with each unit increase of working memory. Performance on the Eyes Test improved each year by about 0.005 points (p < .001). Finally, performance on the Fiction Emotion-Matching task improved by about 0.05 points each year (p = 0.026), with those in the sports condition performing significantly worse over time at a rate about -0.05 less (p = 0.021) than those in the music group.
(DOCX)

## Acknowledgments

We are grateful to Priscilla Perez and Alison Wood for their contributions to this project.

## Author Contributions

**Conceptualization:** Jed Villanueva, Beatriz Ilari, Assal Habibi.

**Data curation:** Jed Villanueva.

**Formal analysis:** Jed Villanueva.

**Writing – original draft:** Jed Villanueva, Assal Habibi.

**Writing – review & editing:** Jed Villanueva, Beatriz Ilari, Assal Habibi.

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
