## [Decision Letter · Decision Letter 0]

21 Nov 2023

PONE-D-23-27763Long-term music instruction partially supports the development of socioemotional skillsPLOS ONE

Dear Dr. Villanueva,

Thank you for submitting your manuscript to PLOS ONE. After careful consideration, we feel that it has merit but does not fully meet PLOS ONE’s publication criteria as it currently stands. Therefore, we invite you to submit a revised version of the manuscript that addresses the points raised during the review process.

The manuscript has been evaluated by three reviewers, and their comments are available below. They have raised a number of minor concerns, could you please carefully revise the manuscript to address all comments raised?

We look forward to receiving your revised manuscript.

Kind regards,

Avanti Dey, PhD

Senior Staff Editor

PLOS ONE

Journal Requirements:

“The Brain and Music Program (AH) at the Brain and Creativity Institute is supported by the GRoW at Annenberg Foundation (https://growannenberg.org/), the Los Angeles Philharmonic Association (https://www.laphil.com/), the Van Otterloo Family Foundation, and the National Endowment for the Arts (grant 1809970-38-18, https://www.arts.gov/). The funders had no role in study design, data collection and analysis, decision to publish, or preparation of the manuscript.”

Reviewers' comments:

Reviewer's Responses to Questions

**Comments to the Author**

1. Is the manuscript technically sound, and do the data support the conclusions?

Reviewer #1: Yes

Reviewer #2: Yes

Reviewer #3: Yes

2. Has the statistical analysis been performed appropriately and rigorously? 

Reviewer #1: Yes

Reviewer #2: Yes

Reviewer #3: I Don't Know

3. Have the authors made all data underlying the findings in their manuscript fully available?

Reviewer #1: Yes

Reviewer #2: Yes

Reviewer #3: Yes

4. Is the manuscript presented in an intelligible fashion and written in standard English?

Reviewer #1: Yes

Reviewer #2: Yes

Reviewer #3: Yes

5. Review Comments to the Author

Reviewer #1: This is an important work. I appreciate the thorough analysis and interpretation.

Line 55: Consider defining what constitutes as “early childhood” for the less informed readers

Line 160: “In this paper, we refer to socioemotional skills as traits and capacities which contribute to an individual’s ability to maintain social relationships and respond to the emotions of others” - Include a citation given the general and/or current terminology misconceptions and variations.

Suggestion: Kuther’s (2020) definition of socioemotional learning and development: “maturation of social and emotional functioning,” which would encompass your more specific definition. A more concise definition would be better, but I have seen this used before in music education literature.

Kuther, T. L. (2020). Livespan development: Livesin context (2nd ed.). SAGE.

Line 205: please clarify “analyzed anonymously”

Line 209: change “is not relationship” to “was no relationship”

Line 222: cite “El Sistema” for readers who are interested in knowing more

Lines 237-243: Consider citing the method of providing stimulus (echoing back patterns) - was it Cohen or Rutkowski?

Lines 237-243: add a small description of the melodies: “5-9 note melodies that were major or minor”

Line 227: omit “a” in “a soccer program” if it was also community-based

Line 268: revise to, “to which participants would then drum along.”

Line 295: revise to, “to which they must respond yes or no.”

Lines 389-400: Please include a sentence as to why d’s were not reported throughout.

Line 440-441: “varied significantly” was confusing as a reader. Please rephrase language to align with other significant findings.

Lines 518-525: Consider making this its own paragraph.

Lines 522-523: This is the first mention that the children played viola and violin. Please include that detail in paragraph lines 220-225

Lines 524-525 implies that the children did not regularly sing in the El Sistema program. Consider mentioning around line 222 whether or not it’s known that students sing regularly, sometimes, or rarely in their program.

Regarding line 550-551: Studies have supported that shorter patterns are less challenging to reproduce for young singers.

See (for example): Sims, W. L., Moore, R. S., & Kuhn, T. L. (1982). Effects of female and male vocal stimuli, tonal pattern length, and age on vocal pitch-matching abilities of young children from England and the United States. Psychology of Music, Spec Its, 104-108.

Nichols, B. E. (2016). Task-Based Variability in Children’s Singing Accuracy. Journal of Research in Music Education, 64(3), 309-321. https://doi.org/10.1177/0022429416666054

Svec, C. L. (2018). The effects of instruction on the singing ability of children ages 5 to 11: A meta-analysis. Psychology of Music, 46(3), 326-339. https://doi.org/10.1177/0305735617709920

Lines 602-607: Overreaching interpretation of the data. Consider omitting and allowing the reader to consider the variety of reasons why children in the sports program shared more than children in the music program (outside the scope of the current paper). INCREDIBLY interesting, however!

Reviewer #2: Please read the reviewer's comments uploaded as a Word document attachment.

The conceptual framework should have been written under specific subtitles that define related terms and concepts of the study.

Besides the aim of your study, please consider writing your research questions explicitly in items. For instance,

1. Research question

2. ……..etc.

Please write this section in subtitles, e.g.

Research instruments

Data collection procedure

Data analysis etc.

Reviewer #3: Dear Jed, Hassal and Beatriz

I enjoyed reading your paper. I can't imagine how much work it should be to lead a longitudinal study, and I am impressed each time I have the opportunity to read your work. What I missed most in reading this article was a clear conceptual framework presented early enough in the text to understand the measured concepts (sharing, empathy, sympathy, theory of mind, etc.). I also wondered about the validity of the tool used to measure sharing behaviors and would have liked to have more information about it. I don't have the necessary knowledge to evaluate your analyses, but I appreciate learning about the results. I also enjoyed the discussion in which you explained them. I wrote some comments, hoping they could help you follow my mind as I was reading the text.

Thanks for the work you are doing.

All the best for the next step,

Take care,

Aimée

6. PLOS authors have the option to publish the peer review history of their article (what does this mean?). If published, this will include your full peer review and any attached files.

Reviewer #1: No

Reviewer #2: No

Reviewer #3: **Yes: **Aimée Gaudette-Leblanc

---

## [Author Response · Author response to Decision Letter 0]

12 Jan 2024

Response to Editor

● We believe the manuscript meets the style requirements for PLOS ONE.

2. If you’ve not already done so, consider depositing your raw data in a repository to ensure your work is read, appreciated and cited by the largest possible audience. 

● Our data has been uploaded to the Open Science Framework and is accessible at https://osf.io/y6dfc/.

3. Thank you for stating in your Funding Statement. Please also include the statement “There was no additional external funding received for this study.” in your updated Funding Statement.

● Our amended Funding Statement is as follows:

The Brain and Music Program (AH) at the Brain and Creativity Institute is supported by the GRoW at Annenberg Foundation (https://growannenberg.org/), the Los Angeles Philharmonic Association (https://www.laphil.com/), the Van Otterloo Family Foundation, and the National Endowment for the Arts (grant 1809970-38-18, https://www.arts.gov/). The funders had no role in study design, data collection and analysis, decision to publish, or preparation of the manuscript. There was no additional external funding received for this study.

4. Please include captions for your Supporting Information files at the end of your manuscript, and update any in-text citations to match accordingly. 

● We have added captions for the supporting information located in the section titled “Supporting information” at the end of the manuscript. 

● We have updated the reference list with additional sources offered by the reviewers. Primarily, we have added references to explain concepts such as El Sistema and psychological constructs such as cognitive and affective empathy, theory of mind, and trait and state empathy. In addition, we have renumbered the in-text citations to match with the new list of references. 

 

Response to Reviewer 1

Line 55: Consider defining what constitutes as “early childhood” for the less informed readers

● We’ve added UNESCO’s definition of early childhood (birth to eight years old).

Line 160: “In this paper, we refer to socioemotional skills as traits and capacities which contribute to an individual’s ability to maintain social relationships and respond to the emotions of others” - Include a citation given the general and/or current terminology misconceptions and variations. Suggestion: Kuther’s (2020) definition of socioemotional learning and development: “maturation of social and emotional functioning,” which would encompass your more specific definition. A more concise definition would be better, but I have seen this used before in music education literature. Kuther, T. L. (2020). Livespan development: Livesin context (2nd ed.). SAGE.

● Using the suggestion provided by the reviewer, we’ve included Kuther’s definition of socioemotional learning and development which we agree encompasses our conceptualization of socioemotional development. 

Line 205: Please clarify “analyzed anonymously”

● We add that to analyze something anonymously means to access the data without any linking information to the participants’ identities. 

Line 209: Change “is not relationship” to “was no relationship”

● Agreed. Text has been altered following the reviewer’s suggestion.

Line 222: cite “El Sistema” for readers who are interested in knowing more

● We’ve added the following additional resources on El Sistema:

Frega, A. L., & Limongi, J. R. (2019). Facts and counterfacts: A semantic and historical overview of El Sistema for the sake of clarification. International Journal of Music Education, 37(4), 561-575.

Majno, M. (2012). From the model of El Sistema in Venezuela to current applications: learning and integration through collective music education. Annals of the New York Academy of Sciences, 1252(1), 56-64.

Uy, M. S. (2012). Venezuela's national music education program El Sistema: Its interactions with society and its participants' engagement in praxis. Music and Arts in Action, 4(1), 5-21.

Lines 237-243: Cite the method of providing stimulus back (Cohen or Rutkowski?)

● We clarify that stimuli were provided back not necessarily following either the Cohen or Rutkowski method. Instead, vocally trained experimenters sang the neutral syllable “la” to the participant, and asked the participant to respond using “la” in return.

Lines 237-243: Add a small description of the melodies (“5-9 note melodies that were major or minor”)

● Agreed. Melodies were clarified to have been 5-9 notes long in either major or minor modes.

Line 227: Omit “a” in “a soccer program” if it was also community-based

● Agreed. Text has been altered following the reviewer’s suggestion.

Line 268: revise to “”to which participants would then drum along.”

● Agreed. Text has been altered following the reviewer’s suggestion.

Line 295: revise to, “to which they must respond yes or no.”

● Agreed. Text has been altered following the reviewer’s suggestion.

Lines 389-400: Please include a sentence as to why d’s were not reported throughout.

● d’s were mistakenly unreported for group comparisons and have now been added throughout.

Line 440-441: “varied significantly” was confusing as a reader. Please rephrase language to align with other significant findings

● We rephrased “varied significantly” to “was significantly predicted by” to follow the reporting of other significant effects.

Lines 518-525: Consider making this its own paragraph

● Agreed, this section has been made its own paragraph. 

Lines 522-523: This is the first mention that the children played viola and violin. Please include that detail in paragraph lines 220-225

● We’ve now included the requested information in the participants subsection.

Lines 524-525 implies that the children did not regularly sing in the El Sistema program. Consider mentioning around line 222 whether or not it’s known that students sing regularly, sometimes, or rarely in the program

● We clarified that students received about 2 hours of choir training each week during their first year.

Regarding lines 550-551: Studies have supported that shorter patterns are less challenging to reproduce for young singer 

● The additional references offered by the reviewer have been incorporated into the paper.

Lines 602-607: Overreaching interpretation of the data. Consider omitting and allowing the reader to consider the variety of reasons why children in the sports program shared more than children in the music program (outside the scope of the current paper). INCREDIBLY interesting, however! 

● Agreed. This section has been removed from the manuscript

Response to Reviewer 2

Too many keywords. Please consider writing specific keywords within the context.

● Agreed. We’ve changed the keywords to the following: El-sistema inspired program, middle childhood, longitudinal, pitch-matching, rhythmic entrainment, socioemotional development 

“Rhythmic” entrainment in keywords 

● Agreed. “Rhythmic” has been added for clarification.

Unclear whether or not systematic instruction in music theory, “technique”, etc. refers to instrumental playing or not

● We’ve added clarifying explaining systematic instruction refers to both instrumental and vocal training.

Line 44: The conceptual framework should have been written under specific subtitles that define related terms and concepts of the study

● We’ve restructured the paper to begin with a section explaining the conceptual framework of the paper. In addition, we’ve added additional subtitles for each primary construct of interest (e.g., Pitch-matching and rhythmic entrainment development, socioemotional development, etc.) 

Lines 188-192: Besides the aim of your study, please consider writing your research questions explicitly in items. For instance, Research question …… etc.

● Agreed. We’ve explicitly included our research questions at the end of the introduction. 

Line 193: Please explain and define the methodology clearly. 

● We believe that the methodology is explained and defined clearly in the following subsections of the Methods section. We choose to include the phrase “Data and code for analysis are available at https://osf.io/y6dfc/” at the beginning of the Methods section following reporting conventions for psychology papers. 

Line 221: “El Sistema” has to be described in concrete 

● Agreed. Additional resources and information has been provided explaining the El Sistema approach.

Line 235: Please write this section in subtitles, e.g. Research instruments, Data collection procedure, Data analysis, etc.

● We have reworded our current subtitles “Measures” and “Procedure” to “Research instruments” and “Data collection procedure”, respectively.

Line 338: Data analysis should be described in the methodology section. In this section, consider writing the research outcomes of the analysis.

● We’ve restructured the paper so that the “Data analysis” section precedes “Results”.

Lines 705: Must be written in APA style, last version.

● We disagree and believe the citation conventions of PLOS ONE follow the Vancouver style.

Response to Reviewer 3

Line 23: What is the aim of this study ? What are the research questions? I would have appreciated a better understanding of the study's aims before learning about the population and the measurement tools used.

● Agreed. We include that our study aims to investigate the development of pitch-matching, rhythmic entrainment, and socioemotional development. We then continue explaining the study population and clarify the socioemotional skills of interest.

Line 111: I would have been interested to know the age of the participants in this study (as you presented the age of the children participating in the two studies cited above).

● We’ve added that Hove and Risen’s (2009) population included college-aged participants.

Line 118: In what context?

● We’ve added additional information explaining that Kirschner and Ilari’s (2014) spontaneous helping involved an experimenter accidentally dropping items and awaited a response from the participant. We’ve also included that the sharing task involved the experimenter implying that they’d like the participants to share their resources during a game.

Line 123: socioemotional skills such as…

● We’ve included “sharing and emotion-recognition” as socioemotional skills of interest potentially related to pitch-matching and rhythmic entrainment.

Line 153: Here, I wonder how old the population being studied in your study is (it was not clearly presented in the abstract... would it be a good idea to do so)?

● We’ve reworded the abstract to more explicitly state that the population was approximately 6.81 years old at baseline data collection.

Line 157: In this study, Brazilian children shared significantly fewer cookies with E2 than Brazilian children. E2 was from Germany. Does culture (or familiarity) have to be considered when studying prosocial behaviors? Would it be relevant to address this point when comparing those two studies?

● Although we agree that culture is important to consider when studying prosocial behaviors, we can’t be certain that the inconsistent findings between Study 1 and Study 2 are a result of culture and choose not to make that claim. The populations in the two studies are not directly comparable, as Study 2 is specifically noted to be highly diverse and not a single homogenous group. As such, the incongruent culture effect seen in Study 1 does not necessarily apply to Study 2 and would require additional research. However, we add that culture is an important contextual factor to consider.

Lines 159-161: I feel a bit lost ... I think that this section might have been presented earlier... Would it be relevant to present a conceptual framework in a section of the text? It would be necessary to refer to texts (articles, chapters) you have read to define this concept.

● Agreed. We’ve introduced a conceptual framework section at the beginning of the introduction to explain our constructs of interest more thoroughly. 

Lines 163-165: Reference?

● We’ve added the following references for theory of mind, affective and cognitive empathy, and empathic responding, respectively:

Premack, D., & Woodruff, G. (1978). Does the chimpanzee have a theory of mind?. Behavioral and Brain Sciences, 1(4), 515-526.

Chakrabarti, B., & Baron-Cohen, S. (2006). Empathizing: neurocognitive developmental mechanisms and individual differences. Progress in Brain Research, 156, 403-417.

Stern, J. A., Botdorf, M., Cassidy, J., & Riggins, T. (2019). Empathic responding and hippocampal volume in young children (Vol. 55, No. 9, p. 1908). American Psychological Association.

Line 165: Add references

● We’ve added the following reference whose definition for socioemotional development we are following as a larger framework:

Kuther, T. L. (2022). The Essentials of Lifespan Development: Lives in Context. SAGE Publications.

Lines 168 and 174: This is not what you are measuring….

● Although we did not measure self-esteem in our study, we believe that it is an important construct to consider when relating socioemotional development and music training. We do, however, add and acknowledge that self-esteem is not measured in our study. 

Lines 177-178: Would it be possible to be precise? Was this study about music education and prosociality - sharing or helping behaviors? This is what you are measuring…

● We add that Rickard, Bambrick, and Gill (2012) measured self-esteem, attitude towards school, social skills, depression, and more. Although not directly related to our study, we again, believe this history of investigating socioemotional skills and music education is important to consider. 

Lines 188-190: This should be in the abstract.

● Agreed. This has been added to the abstract.

Line 196: How old were they at the moment of the data collection used in this study?

● We add clarifying language to state that participants were between 5.73 and 8.01 years old at baseline data collection.

Lines 198-200: Information presented in this sentence doesn't seem related. Could you present them independently?

● Agreed. The sentence has been separated and presented independently. 

Line 219: During those years, did some of the children in the passive control group engage in sports programs? How did you know it?

● For all participants, we were aware of any additional extracurricular activities they engaged in through extensive parental interviews. As such, we know whether children in the passive control group engaged in sports or music programs, and if they did, discontinued participation in the study. 

Line 232: N = ?

● N = 13 has been added

Line 233: N = ?

● N = 15 has been added

Line 255: Could you remind us what was the mean age of the participants at this time of the study?

● We believe that the mean age of participants was clearly stated in the methods section. 

Line 257: Here again, I wonder what age are the children.

● We again believe that the participants’ ages were clearly stated in the methods section

Line 276: Why and how did you adapt the "dictator game"? How did you measure the validity of this tool following the adaptation? 

● We added additional information explaining that this variation of the dictator game was previously used and validated by the following study:

● Flook L, Goldberg SB, Pinger L, Davidson RJ. Promoting prosocial behavior and self-regulatory skills in preschool children through a mindfulness-based Kindness Curriculum. Dev Psychol. 2015 Jan;51(1):44-51. doi: 10.1037/a0038256.

Line 289: This variable is not clearly defined previously.

● Agreed. We have now defined “empathic responding” in the introduction as the capacity to understan

---

## [Decision Letter · Decision Letter 1]

8 May 2024

PONE-D-23-27763R1Long-term music instruction partially supports the development of socioemotional skillsPLOS ONE

Dear Dr. Villanueva,

Thank you for submitting your manuscript to PLOS ONE. After careful consideration, we feel that it has merit but does not fully meet PLOS ONE’s publication criteria as it currently stands. Therefore, we invite you to submit a revised version of the manuscript that addresses the points raised during the review process.

We look forward to receiving your revised manuscript.

Kind regards,

Vanessa Carels

Staff Editor

PLOS ONE

Journal Requirements:

Additional Editor Comments:

We note that one or more reviewers has recommended that you cite specific previously published works. As always, we recommend that you please review and evaluate the requested works to determine whether they are relevant and should be cited. It is not a requirement to cite these works. We appreciate your attention to this request.

Reviewers' comments:

Reviewer's Responses to Questions

**Comments to the Author**

1. If the authors have adequately addressed your comments raised in a previous round of review and you feel that this manuscript is now acceptable for publication, you may indicate that here to bypass the “Comments to the Author” section, enter your conflict of interest statement in the “Confidential to Editor” section, and submit your "Accept" recommendation.

Reviewer #1: All comments have been addressed

Reviewer #4: (No Response)

2. Is the manuscript technically sound, and do the data support the conclusions?

Reviewer #1: Yes

Reviewer #4: No

3. Has the statistical analysis been performed appropriately and rigorously? 

Reviewer #1: Yes

Reviewer #4: Yes

4. Have the authors made all data underlying the findings in their manuscript fully available?

Reviewer #1: Yes

Reviewer #4: Yes

5. Is the manuscript presented in an intelligible fashion and written in standard English?

Reviewer #1: Yes

Reviewer #4: Yes

6. Review Comments to the Author

Reviewer #1: This paper was in good shape through the last revise/resubmit. All requirements above had been met.

Reviewer #4: The authors have addressed most of the detailed comments and suggestions raised by the reviewers of the first submission. I will only focus on major issues that in my view deserve more emphasis and discussion:

- It is difficult to draw conclusions regarding music training effects from this study because children self-selected into training (i.e., predispositions are likely to have played a role), and the music training program was more intense (with more hours per week) than the control training.

- The title suggests a causal effect of music training on socioemotional skills that is not supported by the data. If I read the results correctly, for no measure the music group improved more over time than the controls groups. In fact, the opposite was found for sharing.

- The Abstract needs a conclusion.

- There are several missing meta-analyses and reviews of the literature that are directly relevant for the framing and discussion of this study. For example:

https://link.springer.com/article/10.3758/s13421-020-01060-2

https://www.sciencedirect.com/science/article/pii/S1747938X22000057

https://www.annualreviews.org/content/journals/10.1146/annurev-psych-032323-051354

https://journals.sagepub.com/doi/full/10.1177/17540739211022035

https://www.sciencedirect.com/science/article/pii/S1747938X16300641

7. PLOS authors have the option to publish the peer review history of their article (what does this mean?). If published, this will include your full peer review and any attached files.

Reviewer #1: No

Reviewer #4: No

---

## [Author Response · Author response to Decision Letter 1]

19 Jun 2024

Response to Editor

We have evaluated the works requested by reviewer 4 and have incorporated them into our manuscript.

Response to Reviewer 4

It is difficult to draw conclusions regarding music training effects from this study because children self-selected into training (i.e., predispositions are likely to have played a role), and the music training program was more intense (with more hours per week) than the control training.

• We agree that the interpretation for this study is difficult given selection bias and genetic predispositions. To remedy this, we re-emphasized the potential presence for these two in the discussion.

The title suggests a causal effect of music training on socioemotional skills that is not supported by the data. If I read the results correctly, for no measure the music group improved more over time than the controls groups. In fact, the opposite was found for sharing.

• We modified the title to clarify that the results of this study are not causal but represent associations. Nonetheless, the music group did improve in their emotion-matching ability relative to the sports group, so this point was made clearer in the results and discussion. 

The Abstract needs a conclusion.

• The abstract was modified to include a conclusion and fit within the 300-word limit

There are several missing meta-analyses and reviews of the literature that are directly relevant for the framing and discussion of this study

• We’ve incorporated the recommended articles into the introduction and discussion, briefly covering music training’s association and lack thereof with transfer effects.

---

## [Decision Letter · Decision Letter 2]

4 Jul 2024

Long-term music instruction is partially associated with the development of socioemotional skills

PONE-D-23-27763R2

Dear Dr. Villanueva,

We’re pleased to inform you that your manuscript has been judged scientifically suitable for publication and will be formally accepted for publication once it meets all outstanding technical requirements.

Kind regards,

Laura Morett

Academic Editor

PLOS ONE

Additional Editor Comments (optional):

The most recent version of the manuscript addresses R4's remaining comments sufficiently. Therefore, I am pleased to accept the manuscript for publication in PLOS One.

Reviewers' comments:

Reviewer's Responses to Questions

**Comments to the Author**

1. If the authors have adequately addressed your comments raised in a previous round of review and you feel that this manuscript is now acceptable for publication, you may indicate that here to bypass the “Comments to the Author” section, enter your conflict of interest statement in the “Confidential to Editor” section, and submit your "Accept" recommendation.

Reviewer #4: All comments have been addressed

2. Is the manuscript technically sound, and do the data support the conclusions?

Reviewer #4: Yes

3. Has the statistical analysis been performed appropriately and rigorously? 

Reviewer #4: Yes

4. Have the authors made all data underlying the findings in their manuscript fully available?

Reviewer #4: Yes

5. Is the manuscript presented in an intelligible fashion and written in standard English?

Reviewer #4: Yes

6. Review Comments to the Author

Reviewer #4: The revised version of the manuscript is more accurate and the authors have addressed all my comments.

7. PLOS authors have the option to publish the peer review history of their article (what does this mean?). If published, this will include your full peer review and any attached files.

Reviewer #4: No

---

## [Editor Report · Acceptance letter]

9 Jul 2024

PONE-D-23-27763R2 

PLOS ONE

Dear Dr. Villanueva, 

I'm pleased to inform you that your manuscript has been deemed suitable for publication in PLOS ONE. Congratulations! Your manuscript is now being handed over to our production team.

Kind regards, 

on behalf of

Dr. Laura Morett 

Academic Editor

PLOS ONE